# An Efficiency Coarse-to-Fine Segmentation Framework for Abdominal Organs Segmentation

Cancan Chen, Weixin Xu, and Rongguo Zhang

Infervision Advanced Research Institute, Beijing, China
`{ccancan,xweixin,zrongguo}@infervision.com`

**Abstract.** U-Net has been proved as the most successful segmentation architecture for medical image processing in recent years. Based on this, ResUNet imported ResBlock with skip connection focuses more on the contextual information. In this work, we adopt the 3D ResUNet to build a whole-volume-based coarse-to-fine segmentation framework for the abdominal multi-organs segmentation task, and the mean Dice Similarity Coefficient (DSC) of the segmentation results has achieved 87.67%, the mean Normalized Surface Dice (NSD) has achieved 93.16% on the FLARE2022 validation set. Besides, for each case on the FLARE2022 validation set, the average running time is 19.5614 seconds, and the max gpu memory consumption is 2657 MB.

**Keywords:** Abdominal Organs · Segmentation · FLARE.

## 1 Introduction

Abdominal organ segmentation plays an important role in clinical practice. In recent years, with the development of deep learning, many methods have been proposed to accomplish the segmentation task automatically. In this paper, we focus on multi-organ segmentation from abdominal CT scans. According to the Fast and Low GPU Memory Abdominal Organ Segmentation challenge which required develop segmentation methods that can segment 13 kinds of abdominal organs like the liver, kidney, spleen, pancreas, aorta, IVC, adrenal glands, gallbladder, esophagus, stomach and duodenum simultaneously, we attempted to design our method based on the original ResUNet [5].

In this paper, based on the original ResUNet, we propose a whole-volume-based coarse-to-fine framework. In the first stage, i.e., coarse segmentation, we directly use whole volume CT images and resample it to $128 \times 128 \times 128$ as the input. In the fine stage, 13 organs are split into 2 groups: big organs and small organs. For the big organs, we crop the areas containing the organs based on the coarse segmentation results, and resample the cropped volumes to $160 \times 160 \times 160$ as fine stage input. For the other group, volumes are cropped to $64 \times 256 \times 256$ or $128 \times 128 \times 128$. Specifically, LAG, RAG, Gallbladder and Esophagus are regarded as small organs. Backbones for both stages are 3D ResUNet with 4 down-sample layers encoder and 4 up-sample layers decoders. Notably, the ASPP [3] module

maybe be used to supply the info-loss caused by the multi down-sample on top level.

The main contributions of this work are summarized as follows:

- We propose a whole-volume-based coarse-to-fine framework, which can effectively complete abdominal organs segmentation.
- Based on our proposed framework, we fully utilize the relative position information between big organs by group neighbour organs, which can better locate and segment these organs, especially for stomach, pancreas, duodenum and oesophagus.
- We evaluate our proposed framework on FLARE2022 challenge dataset. The effectiveness and efficiency can be well demonstrated.

## 2   Method

Our proposed method is a whole-volume-based coarse-to-fine framework. Details about the method are described as follows.

### 2.1   Preprocessing

Our proposed method includes the following preprocessing steps:

- Reorientation image to target direction.
- Cropping strategy: None
- Resampling method for anisotropic data:
  Constrained by hardware conditions, the original images are resampled to $128 \times 128 \times 128$ for both coarse segmentation and small organs' fine segmentation task. For the fine segmentation of big organs, images are resampled to $160 \times 160 \times 160$.
- Intensity normalization method:
  Considering that volumes from different centers have different HU values, and this phenomenon appears on the different organs. Therefore, images are clipped to range [-100, 300] and normalized to range [0, 1].
- Others:
  To improve the training and testing efficiency, mixed precision is adopted in the whole process of our framework working.

### 2.2   Proposed Method

The process of our framework is shown in Figure 1. In our proposed framework, coarse segmentation always leads to the error location of small organs, so the 3D ResUNet is cascaded to realize the relocation of small organs. And 8 large organs are divided into 3 groups since that more relative position information can be captured. Figure 2 illustrates the applied 3D ResUNet [5], where a U-Shape architecture is adopted.

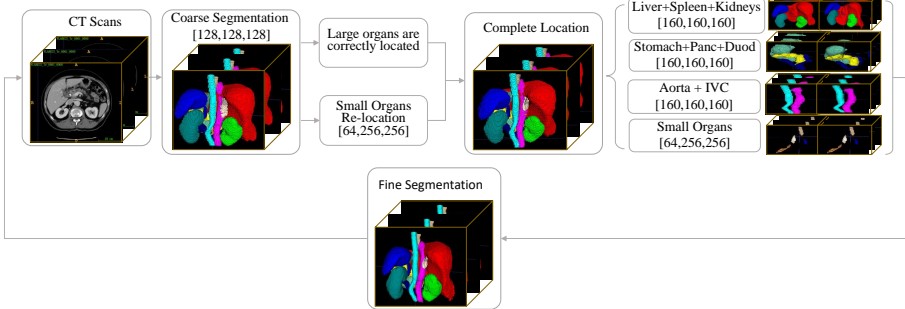

**Fig. 1.** Process of our proposed framework.

Network architecture details: our proposed method is a whole-volume-based coarse-to-fine segmentation framework. For both coarse and fine segmentation stages, the network consists of 4 down-sample layers, 4 up-sample layers for the final segmentation results, and ASPP module.

Loss function: we use the summation between Dice loss and Cross-Entropy loss because compound loss functions have been proved to be robust in various medical image segmentation tasks [9].

### 2.3   Post-processing

To avoid the impact of noise, the connected component analysis [13] is used, and we choose the maximum connected component as the final segmentation results.

## 3   Experiments

### 3.1   Dataset and evaluation measures

The FLARE2022 dataset is curated from more than 20 medical groups under the license permission, including MSD [12], KiTS [6,7], AbdomenCT-1K [10], and TCIA [4]. The training set includes 50 labelled CT scans with pancreas disease and 2000 unlabelled CT scans with liver, kidney, spleen, or pancreas diseases. The validation set includes 50 CT scans with liver, kidney, spleen, or pancreas diseases. The testing set includes 200 CT scans where 100 cases has liver, kidney, spleen, or pancreas diseases and the other 100 cases has uterine corpus endometrial, urothelial bladder, stomach, sarcomas, or ovarian diseases. All the CT scans only have image information and the center information is not available.

The evaluation measures consist of two accuracy measures: Dice Similarity Coefficient (DSC) and Normalized Surface Dice (NSD), and three running efficiency measures: running time, area under GPU memory-time curve, and area

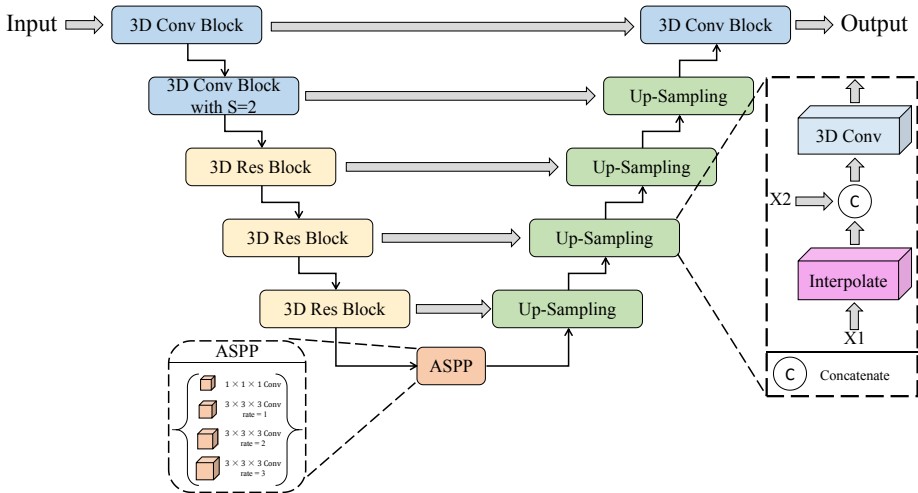

**Fig. 2.** Our proposed network architecture. For the Res Block layer, the stride of the final $1 \times 1 \times 1$ conv is set as 2, that's how we downsample the volumes.

under CPU utilization-time curve. All measures will be used to compute the ranking. Moreover, the GPU memory consumption has a 2 GB tolerance.

### 3.2  Implementation details

**Environment settings** The development environments and requirements are presented in Table 1.

**Table 1.** Development environments and requirements.

| | |
|---|---|
| Windows/Ubuntu version | Ubuntu 18.04.06 LTS |
| CPU | Intel(R) Xeon(R) Silver 4210R CPU @ 2.40GHz |
| RAM | 4×32GB; 3200MT/s |
| GPU (number and type) | Four NVIDIA RTX A6000 48G |
| CUDA version | 11.4 |
| Programming language | Python 3.7 |
| Deep learning framework | Pytorch (Torch 1.7.1+cu110, torchvision 0.8.2) |
| Specific dependencies | |
| (Optional) Link to code | |

**Training protocols** In our training process, we performed the following data augmentation with project MONAI [11] : 1). randomly crop the volumes from range -0.5 to 0.5; 2). add brightness and contrast on the volumes from range

-0.4 to 0.4. 3). random elastic transform with prob=0.5 with sigma from range 3 to 5 and magnitude from range 100 to 200; 4). clip volumes from range 0 to 1. Details of our training protocols are shown in Table 2 and Table 3.

Different organs combination will bring different localization information, which is helpful for fine segmentation. Therefore, we divided the large organs into three groups: (liver, spleen, left and right kidneys); (stomach, pancreas and duodenum); (aorta and IVC). The other four small organs (LAG, RAG, gall-bladder, esophagus) are localized by the 3D ResUNet firstly and then refined by another 3D ResUNet respectively.

**Table 2.** Training protocols.

| | |
|---|---|
| Network initialization | "he" normal initialization |
| Batch size | 8 |
| Patch size | 128×128×128 |
| Total epochs | 100 |
| Optimizer | ADAMW [8] ($weightdecay = 1e - 4$) |
| Initial learning rate (lr) | 1e-4 |
| Lr decay schedule | halved by 20 epochs |
| Training time | 18 hours |
| Loss Function | Summation of Cross Entropy and Dice loss |
| Number of model parameters | 15.17M |
| Number of flops | 93.43G |

**Table 3.** Training protocols for the refine model.

| | |
|---|---|
| Network initialization | "he" normal initialization |
| Batch size | 8 |
| Patch size | 160×160×160 |
| Total epochs | 120 |
| Optimizer | ADAMW [8] ($weightdecay = 1e - 4$) |
| Initial learning rate (lr) | 1e-4 |
| Lr decay schedule | halved by 20 epochs |
| Training time | 18 hours |
| Loss Function | Summation of Cross Entropy and Dice loss |
| Number of model parameters | 15.17M |
| Number of flops | 182.49G |

## 4    Results and discussion

### 4.1    Quantitative results on validation set

As shown in Table 4, our proposed method has achieved mean DSC as 0.8767 and mean NSD as 0.9316 on validation set. The segmentation performance is quite well, especially for the organs with big-size, such as liver, spleen, aorta and kidneys.

**With Unlabelled data.**   In the process of our experiments, we tried to training our model by self-supervised learning with those unlabelled 2000 cases. In summary, we have tried classical methods like MOCOV2 [1], SimSam [2], etc. We also tried to random crop from the original volumes and segment the unlabelled cases by our trained coarse segmentation network and then masked RAG by the segmentation results, then using pix2pix GAN to restore the original volumes, the generator of the pix2pix GAN is utilized as pretrained model. However, all these methods have little effect, and have consumed us much time to attempt these methods. The best mean DSC value on validation set derived from our method, masked RAG and then utilized generator of the pix2pix GAN as pretrained model, with unlabelled data is 0.8416. Moreover, we have tried to combined predicted pseudo label from unlabeled 2000 images with the labeled 50 images, results show that pseudo labels are helpful.

**Table 4.** Results of our proposed method on validation set.

| Organ | DSC | NSD |
|---|---|---|
| Liver | $0.9695 \pm 0.0211$ | $0.9832 \pm 0.0350$ |
| RK | $0.9203 \pm 0.1899$ | $0.9448 \pm 0.1956$ |
| Spleen | $0.9420 \pm 0.0197$ | $0.9807 \pm 0.0421$ |
| Pancreas | $0.8711 \pm 0.0459$ | $0.9643 \pm 0.0442$ |
| Aorta | $0.9426 \pm 0.0248$ | $0.9813 \pm 0.0312$ |
| IVC | $0.9049 \pm 0.0825$ | $0.9272 \pm 0.0885$ |
| RAG | $0.7703 \pm 0.2240$ | $0.8807 \pm 0.2313$ |
| LAG | $0.8009 \pm 0.2352$ | $0.8948 \pm 0.2315$ |
| Gallbladder | $0.8070 \pm 0.2417$ | $0.7943 \pm 0.2475$ |
| Esophagus | $0.8482 \pm 0.1205$ | $0.9315 \pm 0.1207$ |
| Stomach | $0.9122 \pm 0.1430$ | $0.9553 \pm 0.1245$ |
| Duodenum | $0.8102 \pm 0.1023$ | $0.9424 \pm 0.0611$ |
| LK | $0.8977 \pm 0.1746$ | $0.9302 \pm 0.1799$ |
| Average | $0.8767 \pm 0.1250$ | $0.9316 \pm 0.1256$ |

### 4.2   Qualitative results on validation set

Figure 3 shows some failed and successful examples on validation set. It can be found that our proposed method cannot segment gallbladder well on case #2 since that size of gallbladder on this case is too small. Besides, for case #3, because that the stomach is squeezed and displaced, in this caes, stomach was mistakenly segmented as esophagus. Moreover, in case #31, some lesions like tumors in liver may look like gallbladder, this also will influence gallbladder segmentation performance. In case #6, 8 and 35, no lesion in volumes look like neighbour organs, sizes of organs are normal, therefore organs in abdominal can achieve satisfactory segmentation performance.

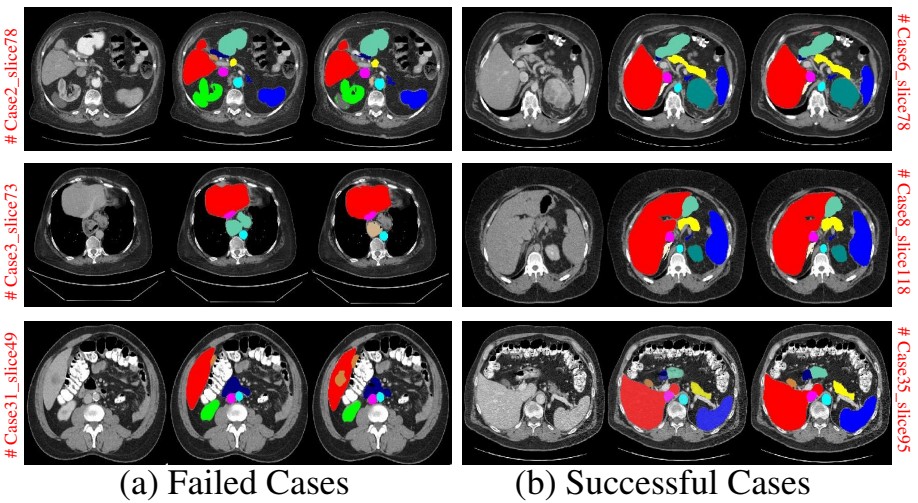

(a) Failed Cases                    (b) Successful Cases

**Fig. 3.** Some failed and successful examples. Columns from left to right are original volumes, ground truth and our predicted results, respectively.

### 4.3   Segmentation efficiency results on validation set

Our segmentation efficiency results on the validation set is shown in Table 5. The average running time of each case in validation set is 19.5614 seconds, the max gpu memory consumption is 2657 MB. The total AUC of GPU time and CPU time are 1,597,650 and 18,215.44, respectively.

### 4.4   Results on final testing set

Our test phase results are shown in table 6, the average DSC and NSD value of 13 organs is 0.8774 and 0.9358, respectively.

**Table 5.** Segmentation efficiency results of our proposed method on validation set.

| Average Running Time | Max Gpu Memory | AUC of GPU Time | AUC of CPU Time |
|:---:|:---:|:---:|:---:|
| 19.5614 Seconds | 2657 MB | 1,597,650 | 18,215.44 |

**Table 6.** Results of our proposed method on test set.

| Organ | DSC | NSD |
|:---:|:---:|:---:|
| Liver | $0.9722 \pm 0.0105$ | $0.9893 \pm 0.0181$ |
| RK | $0.8979 \pm 0.2279$ | $0.9240 \pm 0.2359$ |
| Spleen | $0.9175 \pm 0.1520$ | $0.9587 \pm 0.1616$ |
| Pancreas | $0.8394 \pm 0.0866$ | $0.9485 \pm 0.0886$ |
| Aorta | $0.9218 \pm 0.0638$ | $0.9621 \pm 0.0632$ |
| IVC | $0.9099 \pm 0.0777$ | $0.9400 \pm 0.0919$ |
| RAG | $0.8471 \pm 0.1237$ | $0.9488 \pm 0.1324$ |
| LAG | $0.8484 \pm 0.1604$ | $0.9429 \pm 0.1646$ |
| Gallbladder | $0.8034 \pm 0.2629$ | $0.7977 \pm 0.2683$ |
| Esophagus | $0.8292 \pm 0.1157$ | $0.9277 \pm 0.1132$ |
| Stomach | $0.9175 \pm 0.0983$ | $0.9553 \pm 0.0964$ |
| Duodenum | $0.8020 \pm 0.1063$ | $0.9376 \pm 0.0955$ |
| LK | $0.9001 \pm 0.1950$ | $0.9338 \pm 0.2017$ |
| Average | $0.8774 \pm 0.1292$ | $0.9358 \pm 0.1331$ |

### 4.5   Limitation and future work

In this paper, the performance of small organ segmentation is still not satisfied. In the future, we will focus on the segmentation of those organs, such as gallbladder and adrenal gland. Moreover, self-supervised learning with unlabelled data will also be considered as our future work, and the careful adjustment will further improve the segmentation performance.

## 5   Conclusion

The proposed method can work well on abdominal organs, especially for the organs with big-size, such as liver, spleen and kidneys. Disappointing performance is obtained for AGs and gallbladder because the blurred edges and small-size.

**Acknowledgements** The authors of this paper declare that the segmentation method they implemented for participation in the FLARE 2022 challenge has not used any pre-trained models nor additional datasets other than those provided by the organizers. The proposed solution is fully automatic without any manual intervention.

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
