# OpenReview forum: "An Efficiency Coarse-to-Fine Segmentation Framework for Abdominal Organs Segmentation"
_MICCAI.org/2022/Challenge/FLARE_

### Official Review · Reviewer_fjm3 · 2022-09-16
**Great work and better if unlabelled data can be more utilized**

**Rating:** 8
**Confidence:** 4

**Review:**

Pros: 1. Great resampling and cropping strategy assures good inference efficiency; 2. Divide organs into different groups fully utilizes the relative position information.

Cons: 1. Unlabelled data is not utilized to enhance the model.

---

> ### Author Response · Authors · 2022-10-20
> **Response**
>
> Thanks for your review and suggestion. We have revised our manuscript and added the method that how we use unlabeled data.

---

### Official Review · Reviewer_svJP · 2022-09-16
**Great work and amazing results on DSC and running time.**

**Rating:** 9
**Confidence:** 4

**Review:**

Pros：
1）In this work, the authors devide the large organs into 3 groups and segment them respectively， which is helpful for fine segmentation
2)They do resamling to reduce the resource consumption and tried pix2pix GAN to use the unlabeled data
3)The average runing time per case is 19.56 seconds which is really fast and the mean DSC is 0.8767 which is quite well.
Cons:
1) There are little details about  how to combine the segmentation results of different networks.

It will be better to give the exactly number of filters in encoder and decoder

---

### Official Review · Reviewer_GGFE · 2022-09-16
**Great structure design for this two-phase network**

**Rating:** 9
**Confidence:** 4

**Review:**

Strength: The authors uses a cascade resunet segmentation network along with the application of ASPP to fulfill the segmentation task. The structure design is quite rational.

Weakness: 1. I noticed that the resample sizes are customized for different objects in this dataset, so the generalizability maybe a pitfall of this model.
2. The authors mentioned that the use of unlabeled data did not bring obvious augmentation, maybe this part should be further ameliorated.

---

### Official Review · Reviewer_bKXa · 2022-09-16
**An efficiency coarse-to-fine segmentation method**

**Rating:** 5
**Confidence:** 4

**Review:**

The main contribution of the article is the construction of a two-stage segmentation framework from coarse to fine segmentation using ResUnet.This paper is suitable in length and clearly structured, and performs better in both qualitative and quantitative results, achieving Fast and Low-resource segmentation.
Here are some suggestis:
Figure 4, for example, may be confusing, as it behaves more like a cycle than a two-stage.
The article uses a fully supervised approach, while the competition focuses on how to use unlabeled data, and the authors do not explicitly use a method for unlabeled images.
The results phase does not add examples of visualizations with and without the use of unlabeled images
The reviewers did not find this work very meaningful in terms of methodology.

---

> ### Author Response · Authors · 2022-10-20
> **Response**
>
> Thanks for your review and suggestion. In our figure, we combine the two-stage together, but in practice, it is added a two-stage framework. We have revised our manuscript and added the method that how we use unlabeled data.

---

### Official Review · Reviewer_cWht · 2022-09-18
**MICCAI-FLARE**

**Rating:** 6
**Confidence:** 4

**Review:**

Advance:
1. 'Fully utilize the relative position information between big organs by group neighbour organs, which can better locate and segment these organs, especially for stomach, pancreas, duodenum and oesophagus' is a good and innovative idea.

Weakness:
1. Lack of comparison of unlabel and label data training in experimental analysis.
2. Lack of improved score for ASPP structures analysis.
3. The 3d structure in the picture flow chart of fig.1 seems to be difficult to understand as an example diagram.

---

### Official Review · Reviewer_FLup · 2022-09-19
**Review of An Efficiency Coarse-to-Fine Segmentation Framework for Abdominal Organs Segmentation**

**Rating:** 7
**Confidence:** 4

**Review:**

The authors give detailed description of their method and conduct fully experiments to achieve promising performance.

---

### Official Review · Reviewer_bWx9 · 2022-09-22
**An Efficiency Coarse-to-Fine Segmentation Framework for Abdominal Organs Segmentation**

**Rating:** 9
**Confidence:** 3

**Review:**

Pros: It is a good insight to treat small and big organs separately, achieving high DSC and NSD scores.
Cons: It will be better if add more explanations in discussion section about the self-supervised learning

---

### Meta-Review · Program_Chairs · 2022-09-28

**Recommendation:** Major Revision
**Confidence:** 5

**Metareview:**

Reviewers raise many concerns and suggestions. Please address all comments in the revised manuscript.